# The Tumor Coagulome as a Transcriptional Target and a Potential Effector of Glucocorticoids in Human Cancers

**DOI:** 10.3390/cancers15051531

**Published:** 2023-02-28

**Authors:** Floriane Racine, Christophe Louandre, Corinne Godin, Baptiste Chatelain, Stefan Prekovic, Wilbert Zwart, Antoine Galmiche, Zuzana Saidak

**Affiliations:** 1UR7516, CHIMERE, Université de Picardie Jules Verne, 80054 Amiens, France; 2Service de Biochimie, Centre de Biologie Humaine, 80054 Amiens, France; 3Netherlands Cancer Institute (NKI), 1066 Amsterdam, The Netherlands; 4Center for Molecular Medicine, University Medical Center Utrecht, 3584 Utrecht, The Netherlands; 5Laboratory of Chemical Biology and Institute for Complex Molecular Systems, Department of Biomedical Engineering, Eindhoven University of Technology, 5612 Eindhoven, The Netherlands

**Keywords:** glucocorticoids, tumor coagulome, tumor microenvironment

## Abstract

**Simple Summary:**

Human tumors often establish a local hypercoagulant state that promotes vascular complications, such as venous thromboembolism. The concept of the tumor «coagulome» refers to the repertoire of tumor-expressed genes that locally regulate coagulation and fibrinolysis. Recent systems studies have helped to define the landscape of the coagulome across the spectrum of human tumors, unveiling its link with the tumor microenvironment. Understanding the key elements that regulate the expression of the coagulome is therefore essential. In this study, we explored the dynamic regulation of the tumor coagulome by glucocorticoids. We found that glucocorticoids regulate the coagulome through a combination of direct transcriptional and indirect effects. We show that this transcriptional regulation applies to human tumors, and we suggest that the direct transcriptional regulation of PAI-1 expression by the glucocorticoid receptor may regulate the tumor microenvironment. The transcriptional regulation of the coagulome by glucocorticoids that we report here may have vascular consequences and may account for some of the effects of glucocorticoids on the tumor microenvironment.

**Abstract:**

Background: The coagulome, defined as the repertoire of genes that locally regulate coagulation and fibrinolysis, is a key determinant of vascular thromboembolic complications of cancer. In addition to vascular complications, the coagulome may also regulate the tumor microenvironment (TME). Glucocorticoids are key hormones that mediate cellular responses to various stresses and exert anti-inflammatory effects. We addressed the effects of glucocorticoids on the coagulome of human tumors by investigating interactions with Oral Squamous Cell Carcinoma, Lung Adenocarcinoma, and Pancreatic Adenocarcinoma tumor types. Methods: We analyzed the regulation of three essential coagulome components, i.e., the tissue factor (TF), urokinase-type plasminogen activator (uPA), and plasminogen activator inhibitor-1 (PAI-1) in cancer cell lines exposed to specific agonists of the glucocorticoid receptor (GR) (dexamethasone and hydrocortisone). We used QPCR, immunoblots, small-interfering RNA, Chromatin immunoprecipitation sequencing (ChIPseq) and genomic data from whole tumor and single-cell analyses. Results: Glucocorticoids modulate the coagulome of cancer cells through a combination of indirect and direct transcriptional effects. Dexamethasone directly increased PAI-1 expression in a GR-dependent manner. We confirmed the relevance of these findings in human tumors, where high GR activity/high *SERPINE1* expression corresponded to a TME enriched in active fibroblasts and with a high TGF-β response. Conclusion: The transcriptional regulation of the coagulome by glucocorticoids that we report may have vascular consequences and account for some of the effects of glucocorticoids on the TME.

## 1. Introduction

Human tumors often establish a local hypercoagulant state that promotes vascular thromboembolic complications. Venous thromboembolism (VTE) in particular, is a major source of mortality and morbidity in cancer patients [1,2,3]. The concept of the tumor « coagulome » refers to the repertoire of tumor-expressed genes that locally regulate coagulation and fibrinolysis [4]. Systems biology approaches, especially based on genomics, have been used to study the coagulome [5,6,7]. We have recently used systems biology to characterize the landscape of the human tumor coagulome and its association with the vascular risk, known to strongly depend on the primary tumor type [7]. Indeed, great differences exist across different tumor types in the expression of *F3* mRNA, which encodes the main activator of coagulation, the tissue factor (TF). *F3* is highly expressed in cancers known to be at a high risk of VTE, such as glioblastoma, primary pancreatic, or lung tumors, confirming the local activation of the coagulation cascade by TF as an essential determinant of the risk of hemostatic complications in cancer [7]. 

Recently, the systems biology approach to study the tumor coagulome has also unveiled its close link with the tumor microenvironment (TME), defined as the heterogeneous collection of cells, extracellular matrix, and secreted factors that are present in tumors [8]. The existence of an active interplay between the coagulome and the TME was especially highlighted in the context of fibrinolysis, the essential plasmin-dependent step that leads to the dismantling of polymerized fibrin [5,7]. Human tumors with high mRNA levels of *PLAU*, which encodes the urokinase-type plasminogen activator (uPA), a key activator of plasminogen, have a dense tumor infiltrate of monocytes–macrophages and express high levels of important regulatory immune molecules, such as the immune checkpoint CD276/B7-H3 [7]. In addition to their well-known contribution to hemostasis, it became clear that certain specific products of the coagulation cascade can interact with cognate receptors present on the surface of specific cell types in the TME [5]. The possibility that the coagulome might regulate different cell types within the TME offers promising opportunities, considering the role that the TME plays during carcinogenesis and in the modulation of tumor response to therapy [5,9].

Of all primary tumor types, Oral Squamous Cell Carcinoma (OSCC) and Pancreatic Adenocarcinoma (PAAD) express the highest levels of *F3*, *PLAU*, and *SERPINE1* mRNA (encoding the uPA inhibitor PAI-1, Plasminogen Activator Inhibitor-1) [6,7]. OSCCs, which represent the most frequent type of head and neck cancers [10], have an interesting characteristic of expressing the regulators of coagulation and fibrinolysis at high levels. This characteristic may potentially explain why OSCCs are not at a high risk of vascular complications outside of the therapeutic context [11]. It also makes these tumors an interesting model for examining the regulation of the tumor coagulome and its impact aside from vascular complications. In this respect, a few recent studies have suggested that biomarkers of coagulation might predict tumor recurrence in patients with OSCC after surgical resection, i.e., in a situation of bleeding-related activation of the coagulation cascade [12,13]. Whether the tumor coagulome is a determinant of the oncological outcome in this context remains unknown, but understanding the regulation of the tumor coagulome seems to be of paramount importance, not only because of vascular complications, but also as a determinant of the TME. 

Glucocorticoids are key hormonal regulators involved in stress response that are produced by the adrenal gland. Their interaction with the Glucocorticoid Receptor (GR), a member of the nuclear receptor superfamily encoded by the Nuclear Receptor subfamily 3 group C member 1 (*NR3C1*) gene, is thought to mediate most of their direct transcriptional effects [14]. Glucocorticoids are also known to exert complex anti-inflammatory and immunosuppressive effects [15]. Compared to cortisol, which is the main endogenous ligand of the GR, synthetic agonists of GR, such as dexamethasone, exhibit a higher affinity and potency at the GR, and they are commonly used clinically for their anti-inflammatory properties [15,16]. In cancer patients, corticoids exert pleiotropic effects, ranging from the regulation of immune and inflammatory cells to modulation of the intermediary metabolism [17,18,19]. In recent studies, the GR was also identified as a direct modulator of oncogenic signaling and tumor progression, with different and sometimes contrasting effects on tumor growth depending on the tumor type and/or stage [20,21,22,23,24,25]. The recent study by Obradovic et al. showed that GR activation was able to directly promote metastasis in a mouse xenograft model of breast cancer [23]. More recently, the activation of the GR was also found to be a potential determinant of the response of cancer cells to chemotherapeutic agents [26,27]. Overall, glucocorticoids are emerging as powerful molecules with complex influences on several important facets of human tumor physiology. In this study, we investigated the effects of glucocorticoids on the cancer coagulome and the potential link with the TME. 

## 2. Materials and Methods

### 2.1. Cell Analyses

Details regarding the provenance of all cell lines used here, the cell culture protocols, and a list of all reagents used in this study are described in the Appendix A or published previously [28]. Uncropped immunoblots are available as a Appendix A.

### 2.2. Quantitative Polymerase Chain Reaction (QPCR)

RNA was extracted using RNeasy minikit (Thermofisher, Waltham, MA, USA) and reverse transcribed using High-Capacity cDNA Reverse Transcription kit (Thermofisher). cDNAs were amplified using the TaqMan Universal PCR Master Mix (4304437, Thermofisher) on an ABI 7900HT Sequence Detection System (Applied Biosystems) with gene-specific probes from Thermofisher: *F3* (Hs01076029_m1), *PLAU* (Hs01547054_m1), and *SERPINE1* (Hs00167155_m1). *GAPDH* was used as reference. 

### 2.3. RNA Interference (RNAi)

Silencer select validated siRNA directed against *NR3C1* (s6186) and control siRNA (ref. 4390844) were purchased from Thermofisher. Cells were transfected with siRNAs using Lipofectamine (Thermofisher, Courtaboeuf, France) and Optimem medium (Gibco by Life Technologies, Thermofisher, Courtaboeuf, France).

### 2.4. RNA-seq Analyses and ChIP-seq Analyses 

Whole-tumor RNA-seq data on 321 OSCC and 186 pancreatic adenocarcinoma (PAAD) tumors from TCGA were retrieved from cBioportal at: http://cbioportal.org (accessed on the 1 July 2022) [29,30]. The GR activity score was calculated for each tumor based on 232 genes either positively or negatively associated with GR activity, as previously reported [25]. The gene expression data and GR Chromatin Immunoprecipitation sequencing (ChIP-seq) results from five human lung cancer cell lines (A549, H2122, H460, H1975, and H1944) treated +/− hydrocortisone (2.75 µM, 8 h) were retrieved from the Gene Expression Omnibus (GSE159546) [27]. The genomic loci for *F3*, *PLAU*, and *SERPINE1* were analyzed using the Integrative Genomics Viewer (Broad Institute) to identify direct GR/gene interactions. Single-cell RNA-seq data for 5902 cells (18 HPV head and neck carcinoma tumors) were obtained from the Gene Expression Omnibus (GSE103322) [31]. This study includes data on 2215 malignant cells, as well as other cells of the TME. For 10 of the 18 tumors in this study, data were available for a sufficiently high number of cells for further analysis. The GR activity score was calculated based on West et al. [25].

### 2.5. NCI-60 Analyses

The NCI-60 database, which contains data on 60 cancer cell lines from 9 types of tumors, was accessed using the CellMiner interface (https://discover.nci.nih.gov/cellminer/) (accessed on the 5 February 2023). Basal mRNA expression levels (Agilent mRNA/log2) were retrieved for the 60 cell lines [32]. Gene expression data from 54 of the cell lines originating from solid tumors were kept for further analysis; the six leukemia cell lines were excluded from analysis.

### 2.6. Pathway Enrichment Analyses 

Gene Set Enrichment Analysis (GSEA) was performed on OSCC and PAAD from TCGA using the Java GSEA desktop application (https://www.gsea-msigdb.org/gsea/index.jsp) (accessed on the 10 September 2022). We used C2 curated gene sets to compute the overrepresentation of specific gene sets in tumors with high expression of *F3*, *PLAU*, or *SERPINE1* (high vs. low expression defined by the median). The analyses were performed using 1000 permutations [33]. Gene Ontology (GO) analyses were performed using PANTHER (Protein ANalysis THrough Evolutionary Relationships, http://www.pantherdb.org/ (accessed on the 4 October 2022)) [34]. Statistical overrepresentation test/GO Biological Process complete analysis was performed using Fisher’s exact test with FDR correction. The comparisons were performed against *Homo sapiens* all genes database.

### 2.7. Tumor Microenvironment Composition Analyses 

The Microenvironment Cell Population counter (MCP counter) analysis was used to quantify the relative abundance of ten types of immune and stromal cell populations based on the RNA-seq data in OSCC and PAAD TCGA tumors [35]. Immune-related scores (TGF-β response) were retrieved from Thorsson et al. [36].

### 2.8. Statistical Analyses 

Student’s *t* test or Wilcoxon–Mann–Whitney test was used to compare continuous variables, and Chi2 test was used to compare categorical data, as indicated. *p* < 0.05 was considered a threshold for significance. A False Discovery Rate (FDR) correction was applied when indicated. All statistical analyses were performed with R version 4.1.0 (https://www.r-project.org (accessed on the 4 October 2022)).

## 3. Results

### 3.1. Transcriptional Regulation of the Coagulome by Glucocorticoids in Cancer Cells 

To evaluate the effect of glucocorticoids on cancer cells, we treated OSCC cell lines PE/CA-PJ34 and PE/CA-PJ41 with dexamethasone in vitro. Long-term treatment with dexamethasone (6 days) had a small effect on cell viability (a 12% decrease for PE/CA-PJ41 and a 37% decrease for PE/CA-PJ34 at the maximal concentration of 10 µM) (Appendix A). We observed a major effect on PAI-1 expression with a significant 5-fold increase after 6 days of treatment of PE/CA-PJ41 cells with dexamethasone (10 µM) (Figure 1A,C; Appendix A). In both cell lines, PAI-1-specific immunofluorescence labeling was homogeneous across individual cells, suggesting that the regulation was not limited to a subpopulation of cancer cells (Appendix A). Dexamethasone also significantly decreased TF and uPA expression in PE/CA-PJ41 (30.7% for TF, *p* = 1.8 × 10^−4^ and 34.2% for uPA, *p* = 2.4 × 10^−3^) after 6 days of treatment. Note that PAI-1 was not detectable in the cell extracts of the PE/CA-PJ34 cell line (Figure 1A). However, PAI-1 was detectable in the supernatants, and we confirmed the increase in PAI-1 expression after dexamethasone treatment (Figure 1B). We examined the possibility of a transcriptional regulation by measuring gene expression by QPCR. We observed that dexamethasone induced a significant increase in the expression of *SERPINE1* mRNA, the gene that encodes PAI-1, in PE/CA-PJ41 (>4-fold increase vs. control after 6-day exposure, *p* < 0.05) (Figure 1D). Dexamethasone treatment also significantly decreased the expression of *F3* (encoding TF) and *PLAU* (encoding uPA) (*p* < 0.05), reaching the lowest levels after 48 h (Figure 1D). The effect of dexamethasone on the expression of *F3* and *PLAU* was apparent from submicromolar concentrations (500 nM), pointing to the specificity of these effects (Appendix A).

To address the mechanism of action of glucocorticoids on cancer cells in an inflammatory context, we examined the effect of dexamethasone in the presence or absence of the cytokine Tumor Necrosis Factor-α (TNF-α). TNF-α applied on PE/CA-PJ34 and PE/CA-PJ41 cells sharply increased *F3* and *PLAU* mRNA and slightly increased *SERPINE1* mRNA expression levels in both cell lines. Dexamethasone prevented the increase in *F3* and *PLAU* in both cell lines (Figure 2). Conversely, dexamethasone directly increased *SERPINE1* mRNA expression independently of TNF-α exposure (Figure 2). These findings were consistent with a dual effect of glucocorticoids on OSCC, with a predominant effect on anti-inflammatory signaling accounting for the negative regulation of TF and uPA, as opposed to a positive regulation of PAI-1.

### 3.2. Role of the GR in the Regulation of PAI-1 Expression

To address the role of the GR in the effects produced by dexamethasone, we used RNA interference directed against the gene *NR3C1* that encodes GR. With this approach, we successfully decreased GR expression by >70% in both PE/CA-PJ34 and PE/CA-PJ41 cell lines (Figure 3A; Appendix A). Our results show that a decreased expression of GR (using siNR3C1) prevented the dexamethasone-induced increase in PAI-1 (Figure 3B; Appendix A). These results were confirmed at the mRNA level, with no increase in *SERPINE1* mRNA with dexamethasone treatment in cells treated with siNR3C1 (Figure 3C). The decreased expression of GR (siNR3C1) did not prevent the decrease in TF and uPA expression with dexamethasone treatment, further confirming our observation regarding the dual regulation of the coagulome (Figure 3B). These findings were supported with a distinct pharmacological approach using a GR antagonist, mifepristone, which also prevented the increase in PAI-1 expression (Appendix A), again strongly suggesting the existence of direct GR-dependent regulation of PAI-1 expression in cancer cells. To address the relevance of these findings in a broader setting, we used data from a previous study (GSE159546) reporting the transcriptome of five lung cancer cell lines treated with hydrocortisone (2.75 µM, 8 h) [27]. Consistent with our in vitro results with OSCC, an induction of *SERPINE1* was apparent in three of the cell lines (A549, H2122, and H1975). A decrease in *PLAU* expression was observed in four out of the five lung cancer cell lines. There was no consistent change in *F3* (Figure 4A). 

In order to directly test the interaction between the GR and these genes, we used ChIP-seq data from GSE159546. In three out of the five human lung cancer cell lines (A549, H2122, and H1975), hydrocortisone stimulated the interaction between GR and exon 9 of the *SERPINE1* gene. For three of the five cell lines, a direct interaction was observed either with the promoter region of *SERPINE1* (A549 and H1944) or exon 1 of the gene (H1975) (Figure 4B). Multiple GR interaction sites were also observed in the 30 kb upstream region of the *SERPINE1* coding sequence (Figure 4B). These data further established the existence of direct genomic effects of GR on the *SERPINE1* locus in cancer cells. 

### 3.3. Transposition to Human Tumors—Microenvironment Characteristics of Tumors with High GR Activity and SERPINE1 Expression

Next, we used a “GR activity score” described previously that is based on a GR activity profile in breast cancer cells [25]. We calculated the “GR activity score” for the 54 cell lines originating from solid tumors from the NCI-60 database. We observed that the GR activity score was positively correlated with *SERPINE1* expression (r = 0.34, *p* = 0.013, Spearman) (Figure 5). Neither *F3* nor *PLAU* expression was significantly correlated with the GR activity score. 

To examine the relevance of glucocorticoid regulation of the coagulome in human tumors, we used whole exome RNA-seq data from 321 OSCC and 186 PAAD tumors retrieved from TCGA (Figure 6). We performed GSEA to compare tumors stratified according to their expression of *F3*, *PLAU*, and *SERPINE1* (by the median). Tumors with high levels of these coagulome genes were positively associated with the gene set “WP Glucocorticoid Receptor Pathway”, with the highest enrichment score for tumors expressing high levels of *SERPINE1* (OSCC: NES = 2, FDR q = 0.009; PAAD: NES = 2.27, FDR q = 0.002) (Figure 6). These results further suggest the existence of a link between glucocorticoid signaling and the expression of *F3*, *PLAU*, and *SERPINE1* in human tumors.

We wanted to address whether tumors with high GR activity, as determined by West et al. [25] (upper half by median), combined with high *SERPINE1* expression (upper half by median), here called “High++” (Figure 7A, Appendix A), have a specific composition of their tumor microenvironment. A comparison of the clinical characteristics of the “High++” OSCC vs. all other OSCC showed that these tumors are of a higher grade (*p* = 0.0023, Chi2). The rates of the angiolymphatic invasion (ALI), extracapsular spread (ECS), and TNM stage were not significantly different between the two groups (Appendix A). We then used the MCP-counter to compare the relative infiltrate of different cell types in “High++” vs. all other OSCC tumors [35]. We found that for most cell types studied, including T cells, B cells, neutrophils, CD8 T cells, cytotoxic lymphocytes, NK cells, and dendritic cells, there was no difference between the “High++” group and the rest of the OSCC tumors. Significantly higher levels of fibroblasts (*p* = 0.0031, FDR), endothelial cells (*p* = 5.29 × 10^−4^, FDR), and cells of the monocytic lineage (*p* = 0.0024, FDR) were present in the “High++” group compared to other OSCCs (Figure 7B). These observations were confirmed in PAAD tumors, with significantly higher infiltration levels of fibroblasts (*p* = 0.02017 FDR), cells of the monocytic lineage (*p* = 0.0162 FDR), and endothelial cells (*p* = 7.33 ×10^−4^ FDR) in tumors with high GR activity score/*SERPINE1* (Appendix A). In PAAD, tumor CD8 T cells (*p* = 0.02113 FDR) and cytotoxic lymphocytes (*p* = 0.0055 FDR) were also enriched in High++ tumors (Appendix A).

After performing a differential gene expression (DEG) analysis, we used the top 200 genes that were the most significantly enriched in the “High++” group in a Gene Ontology study. The GO terms that were enriched to the greatest extent in the “High++” tumors are shown in Figure 7C, and are listed in Appendix A and include, for example, the GO term “wound healing” (Figure 7C). We also examined the immune-related scores defined by Thorsson et al. [36] in relation to the “High++” status of OSCC tumors. The most significant difference was found in the “TGF-β response” (1.75-fold higher TGF-β response in “High++” vs. “other”, *p* = 1.12 × 10^−9^ FDR). The expression of TGFB1, the gene encoding TGF-β, was significantly higher in “High++” tumors compared to the rest (*p* = 0.0013) (Figure 7D).

### 3.4. Single-Cell Analysis of SERPINE1 Regulation by GR

To address the regulation of *SERPINE1* by GR at a higher resolution in human tumors, we used RNA-seq data from a single-cell analysis of OSCC [31]. We calculated the “GR activity score” in the cancer cell population using 163 available genes from the 232 genes that constitute the signature. We analyzed the expression of *SERPINE1* in all cancer cells (*n* = 2215) with either a high or low GR activity score (divided by the median). We found a significantly higher expression of *SERPINE1* in cancer cells with a high GR activity score compared to those with low GR activity (*p* = 1.92 × 10^−8^) (Figure 8A). We identified 10 tumors from GSE103322 that had data on enough cells, i.e., >100 cells, for further analysis. For each of these tumors, we examined the extent to which the GR activity score and *SERPINE1* expression overlapped (Figure 8B). The percentage of cancer cells with high GR activity and high *SERPINE1* expression, as indicated in orange in Figure 8B, was calculated in individual tumors. For the tumor with the highest GR activity score (#17), the overlap with *SERPINE1* expression was at 65.45%, whereas for the tumor with the lowest average GR activity score (#22), only 10% of the cancer cells had both high GR activity and a high expression of *SERPINE1* (Figure 8B). We found a positive correlation (Spearman coefficient r = 0.7, *p* = 0.02) between the GR activity score for each tumor and the percent of cancer cells with *SERPINE1* coexpression. These findings again validate our conclusion regarding the regulation of *SERPINE1* by GR, suggesting that this regulation applies at the single-cell level.

## 4. Discussion

Glucocorticoids are powerful transcriptional regulators with broad effects on tumor physiology [37]. We investigated their effect on the coagulome of different primary cancer types: OSCC, a tumor type characterized by a high expression of TF and uPA [6] and lung and pancreatic cancers, two tumor types known to predispose cancer patients to a high risk of VTE. While previous studies have addressed the role and regulation of uPA and PAI-1, i.e., the core components of the coagulome, their regulation by glucocorticoids had to our knowledge not been shown in these tumors [38]. We observed a dual regulation of the coagulome by glucocorticoids, consisting of an indirect effect, possibly related to the inhibition of inflammatory signaling in cancer cells, and a direct transcriptional effect that applies to PAI-1 (*SERPINE1*). These findings were confirmed in a broad analysis, using a large panel of human cancer cells from the NCI-60 database that include 54 cell lines originating from eight types of primary solid tumors. We addressed the relevance of our findings in human tumors by showing that the expression of the coagulome genes, especially *SERPINE1* (PAI-1), is linked to the “glucocorticoid receptor pathway” in OSCC and PAAD. We examined the possible consequences of this GR-dependent regulation by analyzing the cellular composition of the TME of these tumors. In tumors with high GR activity and high *SERPINE1* expression, the TME was enriched in stromal cells (endothelial cells and fibroblasts) and monocytic cells. Finally, we suggest that the contribution of the GR–*SERPINE1* axis also applies at the single-cell level. Based on our study, we therefore propose that the tumor coagulome is a target of glucocorticoids.

Recent genomic studies have provided a detailed analysis of the landscape of the tumor coagulome across different tumor types, but we still know little about its dynamic regulation [5,7]. Our study further illustrates the complex effects of glucocorticoids on human tumors [37] and highlights a novel facet of their action. Our in vitro and in silico analyses suggest that components of the fibrinolysis cascade (uPA and PAI-1) are potentially more directly and more potently regulated than the coagulation cascade (TF) in human tumors. Importantly, the positive regulation of PAI-1 expression that we observed is consistent with the previously established positive transcriptional regulation of PAI-1 reported in various primary tissues [39,40]. It is also consistent with previous reports of a functional Glucocorticoid Response Element (GRE) in the *SERPINE1* promoter [41]. High fibrinolytic activity, catalyzed by a high expression of uPA, the main upstream regulator of fibrinolysis, has been previously suggested to be a negative determinant of thrombosis risk in human tumors, by promoting the rapid turnover of fibrin clots [11]. Our findings open the interesting possibility that glucocorticoids might change the tumor fibrinolytic activity and therefore possibly increase the risk of VTE. Importantly, predicting the vascular risk of individual cancer patients remains notoriously difficult [3]. Whether the exploration of the tumor glucocorticoid signaling status might provide useful information in this context is a speculative yet promising possibility that remains to be tested.

In addition to its impact on the vascular risk in cancer patients, the coagulation and fibrinolysis cascades potentially regulate tumor progression. Several experimental studies indicate, for example, that the fibrinolysis cascade may regulate the turnover of proteins of the extracellular tumor matrix, with possible consequences on tumor cell invasive growth [38]. While the regulation of the tumor matrix has been the focus of many biochemical studies, the link between the fibrinolysis cascade and the regulation of tumor stromal cells has been less studied, despite the key role played by these cellular components of the TME in tumor growth and response to treatments. With this in mind, a focus of our study was to use genomics to analyze the tumor ecosystem of OSCC and PAAD with high GR activity/high *SERPINE1* expression. We identified fibroblasts, monocytic, and endothelial cells as the cell types with an increased density in the TME of these tumors. Interestingly, these findings are consistent with previous in vitro studies that have reported direct positive effects of PAI-1 on tumor endothelial cells [42], monocytic cells [43,44], and cancer-associated fibroblasts [45,46]. These studies suggest that PAI-1, and perhaps other components of the tumor coagulome, might not only be a target, but also an effector of glucocorticoids in tumor tissues. In this respect, studies performed on PAI-1 KO animals indicate that PAI-1 mediates the effects of glucocorticoids on bone and muscle tissues [47,48,49]. Based on our results and the conclusions of these studies, we speculate that a GR–*SERPINE1* axis might be functional in human tumors. Importantly, a major limitation of our study, as is often the case with genomic studies addressing the regulation of the TME, is the fact that we examined the consequences of the GR–*SERPINE1* axis with correlative analyses. Further studies are therefore warranted to experimentally confirm the consequences of the activation of the GR–*SERPINE1* axis on the TME, including the regulation of the tumor matrix and the regulation of noncancer cells of the tumor stroma. A deeper knowledge of the consequences of GR–*SERPINE1* signaling offers the exciting perspectives of a better understanding of the effects of glucocorticoids on tumor physiology.

## 5. Conclusions

Glucocorticoids modulate the coagulome of cancer cells through a combination of direct transcriptional and indirect regulatory effects. We found that dexamethasone directly increased PAI-1 expression in a GR-dependent manner, and we confirmed the relevance of these observations in different types of tumors. The transcriptional regulation of the coagulome by glucocorticoids that we report here may not only have vascular consequences, but it may also account for some of the effects of glucocorticoids on the TME.

## Figures and Tables

**Figure 1 cancers-15-01531-f001:**
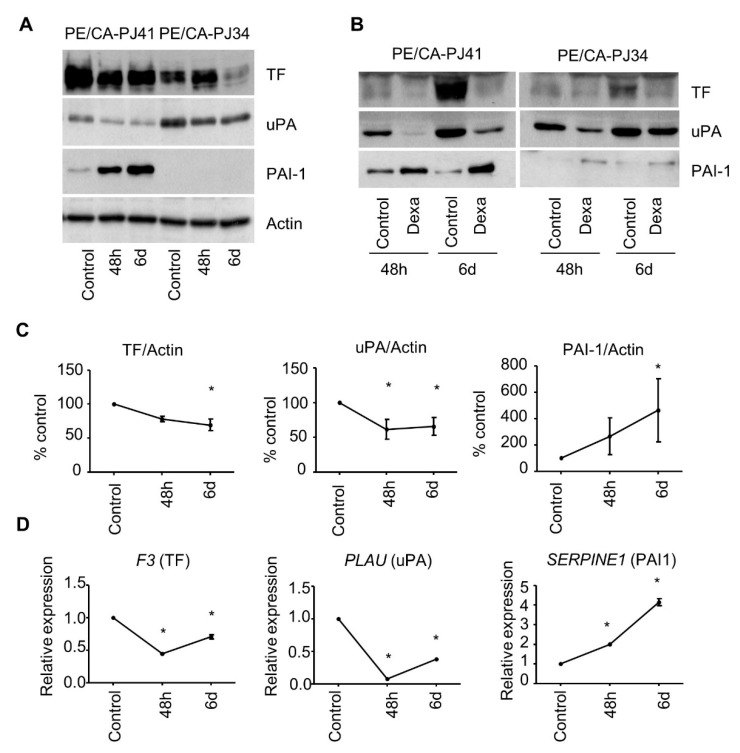
The impact of glucocorticoid treatment on the coagulome of OSCC cells. (**A**) Immunoblot analysis of the expression of TF, uPA, and PAI-1 in the OSCC cell lines PE/CA-PJ41 and PE/CA-PJ34. Cells were treated with dexamethasone (10 µM) for 48 h or 6 days. Actin was used as a loading control. (**B**) Immunoblot analysis of the expression of TF, uPA, and PAI-1 in the supernatants of the OSCC cell lines. (**C**) Quantification of protein expression detected by immunoblot shown in (**B**) for PE/CA-PJ41 using the Image J software. Student’s *t* test was used to compare each condition with control (* *p* < 0.05). (**D**) QPCR analysis of the expression of *F3*, *PLAU*, and *SERPINE1* in PE/CA-PJ41 cells treated with dexamethasone (10 µM) for 48 h or 6 days. Results are represented as relative expression, with control set as 1.

**Figure 2 cancers-15-01531-f002:**
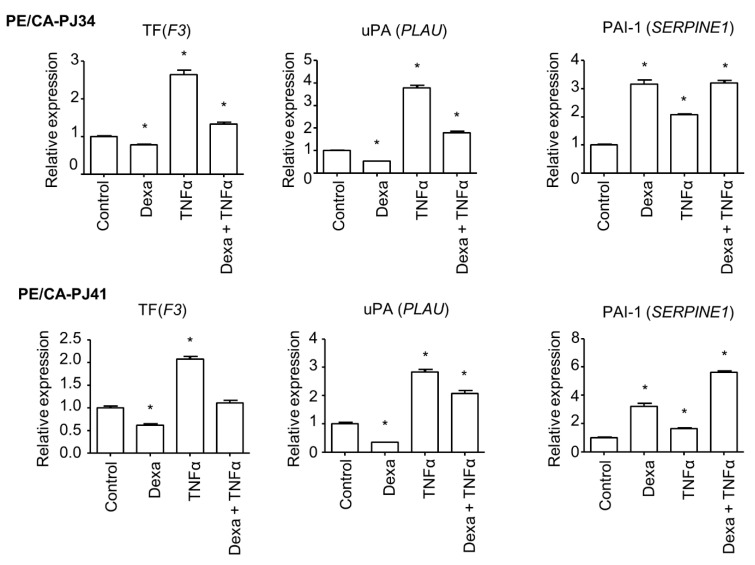
The role of proinflammatory signaling in the regulation of coagulome gene expression. QPCR analysis of the expression of *F3*, *PLAU*, and *SERPINE1* in PE/CA-PJ34 and PE/CA-PJ41 (data from a representative experiment, out of three independent experiments). Cells were pre-exposed to dexamethasone (10 µM) for 1 h before TNFα (25 ng/mL) treatment for 48 h. Student’s *t* test was used to compare each condition with control (* *p* < 0.05).

**Figure 3 cancers-15-01531-f003:**
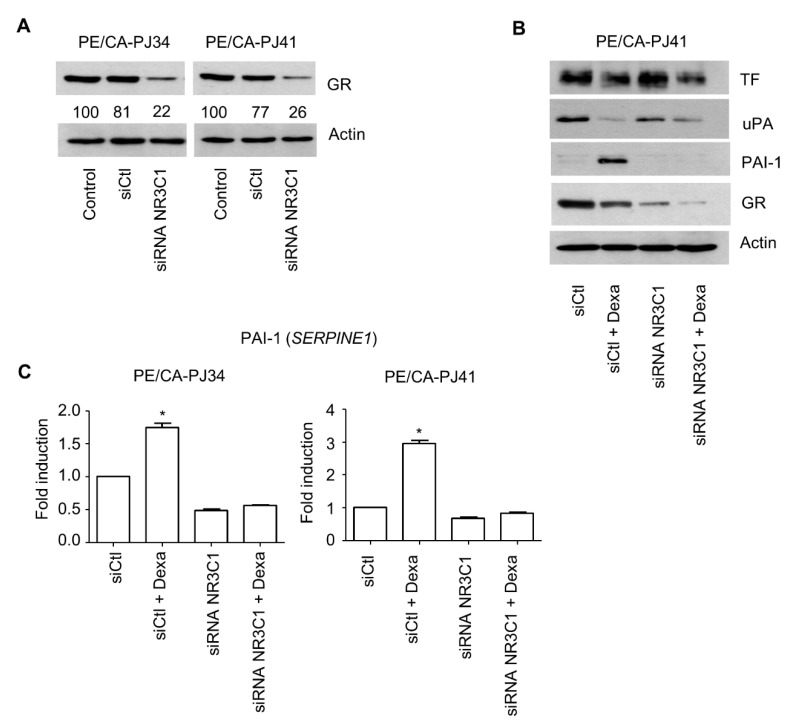
The implication of the glucocorticoid receptor in the regulation of PAI-1. (**A**) Immunoblot analysis of the expression of GR in OSCC cell lines after transfection with siRNA directed against *NR3C1* or siControl. Quantification values are given under the GR immunoblots and were calculated using the Image J software (v1.51j8, National Institutes of Health, Bethesda, MD, USA). (**B**) Immunoblot analysis of the expression of TF, uPA, PAI-1, and GR in PE/CA-PJ41 +/− dexamethasone (10 µM), +/− siRNA NR3C1. Actin was used as a loading control. (**C**) QPCR analysis of the expression of *SERPINE1* in PE/CA-PJ34 and PE/CA-PJ41. Cells were treated with dexamethasone (10 µM) for 48 h, 30 h after siRNA transfection. Student’s *t* test was used to compare each condition with control (* *p* < 0.05).

**Figure 4 cancers-15-01531-f004:**
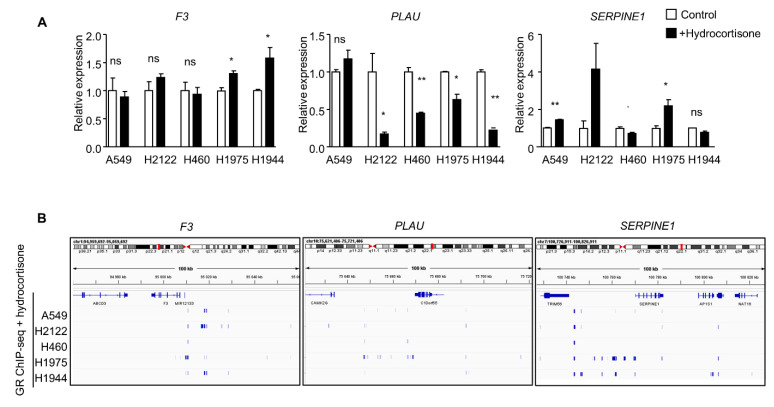
Coagulome gene expression in lung cancer cell lines treated with glucocorticoids. (**A**) Expression of *F3*, *PLAU*, and *SERPINE1* was analyzed in five lung cancer cell lines: A549, H2122, H460, H1975, and H1944, treated with hydrocortisone (2.75 µM) for 8 h. Gene expression was analyzed by mRNA sequencing (data from GSE159546, using raw counts [27]) (* *p* < 0.05; ** *p* < 0.01, ns = not significant). (**B**) GR ChIP-seq analysis of the *F3*, *PLAU*, and *SERPINE1* loci in lung cancer cell lines treated with hydrocortisone (GSE159546).

**Figure 5 cancers-15-01531-f005:**
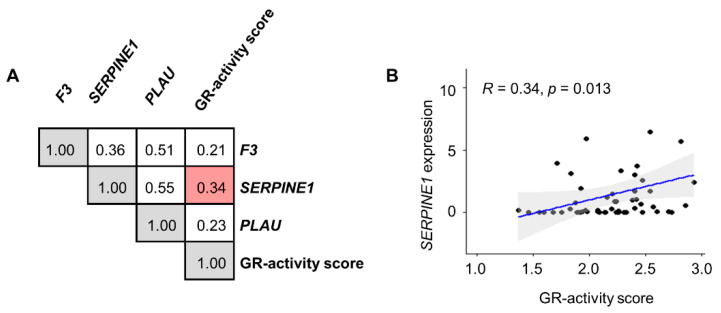
Correlation analysis between the coagulome gene expression and the GR activity score in human cell lines from the NCI-60. (**A**) Correlation matrix showing the Spearman correlation coefficients between *F3*, *PLAU*, and *SERPINE1* expression (NCI-60, basal mRNA expression, and Agilent) and the GR activity score in 54 cell lines from eight different types of solid tumors. (**B**) A correlation plot for *SERPINE1* expression vs. the GR activity score in the 54 solid-tumor cell lines from NCI-60.

**Figure 6 cancers-15-01531-f006:**
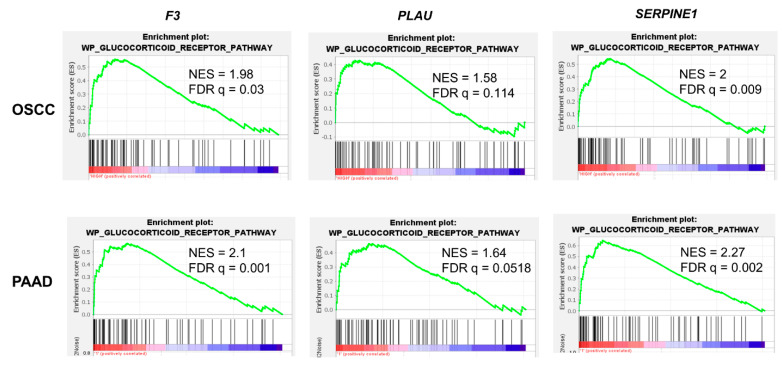
Analysis of the possible physiological link between glucocorticoids and the coagulome. GSEA analysis of OSCC and PAAD tumors according to their expression of the coagulome genes *F3*, *PLAU*, and *SERPINE1*. Tumors with high expression of the coagulome genes had an enrichment in signaling of the “WP_glucocorticoid_receptor_pathway”. NES = normalized enrichment score. A False Discovery Rate correction was applied as indicated.

**Figure 7 cancers-15-01531-f007:**
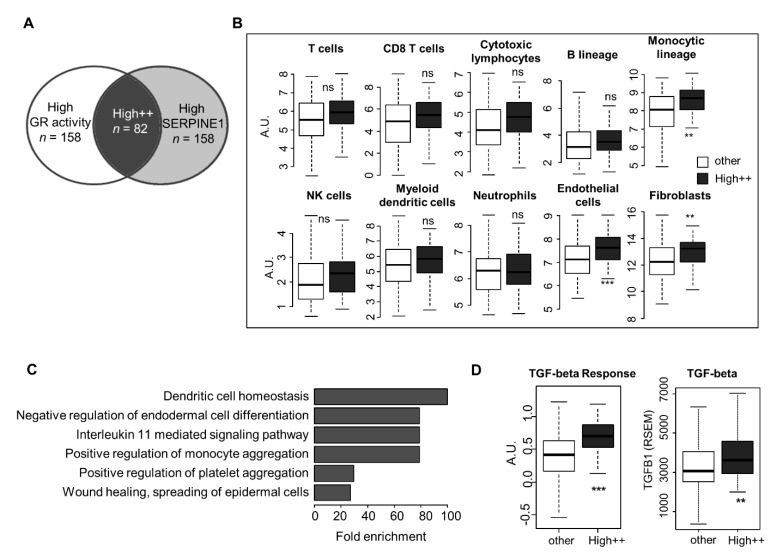
Tumor microenvironment analysis in OSCC tumors with high GR activity and high *SERPINE1* expression. (**A**) Venn diagram showing the overlap between high GR activity and high *SERPINE1* expression (upper half for both) (i.e., “High++”) in OSCC/TCGA tumors, *n* = 82 [25]. (**B**) The abundance of eight types of immune cells and endothelial cells and fibroblasts in OSCC/TCGA tumors stratified according to their “GR activity score” and *SERPINE1* expression. Wilcoxon–Mann–Whitney was used to compare groups. * *p* < 0.05 after FDR was set as threshold for significance (** *p* < 0.01; *** *p* < 0.001, ns = not significant). (**C**) Gene ontology analysis of the top 200 genes most significantly overexpressed in “High++”. (**D**) “TGF-beta response” score in “High++” OSCC tumors vs. other tumors and TGFB1 expression (RSEM) in OSCC tumors from TCGA (** *p* < 0.01; *** *p* < 0.001).

**Figure 8 cancers-15-01531-f008:**
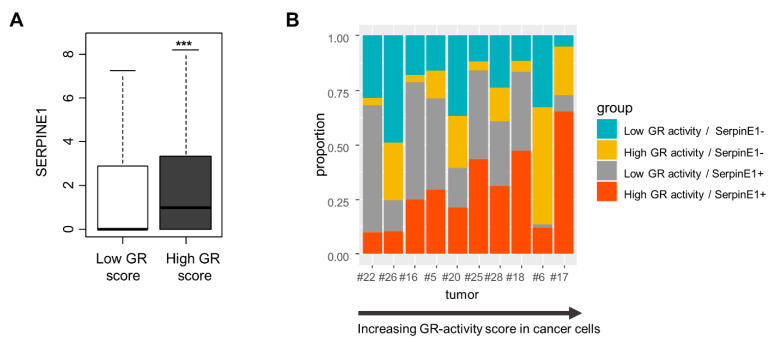
Single-cell analysis of *SERPINE1* regulation by GR. (**A**) Boxplots showing *SERPINE1* expression in single cancer cells of OSCC (GSE103322, *n* = 2215 cells) stratified according to their GR activity score (by median, based on [25] (*** *p* < 0.001). (**B**) Graph showing the overlap between the GR activity score and *SERPINE1* expression in 10 head and neck tumors from the study by Puram et al. [31]. The tumors were organized by ascending “GR activity score”. Orange: high GR activity score/*SERPINE1*+; grey: low GR activity score/*SERPINE1*+; yellow: high GR activity score/*SERPINE1*-; turquoise: low GR activity score/*SERPINE1*-.

## Data Availability

Raw experimental data for the Figures and Appendix A are available upon request. Other data used in this study are publicly accessible (TCGA, GSE103322, GSE159546).

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
