# Peer review of "The Tumor Coagulome as a Transcriptional Target and a Potential Effector of Glucocorticoids in Human Cancers"

_cancers, 2023, doi:10.3390/cancers15051531_

Round 1
Reviewer 1 Report
1. Please provide rationale for using dexamethasone (10µM) in PE/CA-PJ34 and PE/CA-PJ41. Please supplement data with lower and higher concentration respectively. Authors would agree on dose-dependent effect which may modulate results and expression profile respectively.
2. Line 175-176: In both cell lines, PAI-1-specific 174 immunofluorescence labelling was homogeneous across individual cells, suggesting that the regulation was not limited to a subpopulation of cancer cells (data not shown). Its advised to provide immunofluorescence data in supplementary files.
3. Authors report, PAI-1 was not detectable in the cell extracts, however, PAI-1 was detectable in the supernatants and we confirmed the increase in PAI-1 expression after dexamethasone treatment. I would recommend presenting expression data (flow analysis) for PAI-1 with and without dexamethasone. Does it mean PAI-1 is undergoing shedding.
4. Please plot independent replicates in each graph for representation and update the figure 2 and Figure 3C.
5. Manuscript is badly formatted, kind request to review the manuscript for throughout for respective fonts and description of text is inappropriate at many places.
6. I would not agree with manuscript title to study the role in cancer as the data is specifically with respect to OSCC, please update the title with respect to OSCC. Another suggestion is to supplement data with respect to cell lines of thyroid cancer and pancreatic cancer (refer https://www.proteinatlas.org/ENSG00000122861-PLAU/cell+line) with high PLAU expression for the comaprative profile if authors want to float the manuscript broadly with respect to cancers. I would recommend have comparative results from the remaining two cancers as well.
Author Response
Dear Editor, dear Reviewers,
We would like to submit a revised version of our manuscript, entitled «The tumor coagulome as a transcriptional target and a potential effector of glucocorticoids in human cancers», by Racine et al., to « Cancers » (manuscript ref. 2213006).
We are very grateful for the constructive and insightful comments that we have received. Our detailed responses can be found in this letter. Compared to the initial version of the manuscript, some important changes have been introduced:
- A new Fig. 5 and two new Supplementary figures (Suppl. Fig. 2 and 3) have been introduced in order to address some of the questions/requests from our reviewers. The new data further support the broad relevance of our conclusions to different types of human tumors.
- Great effort was made to expand the introduction, insert a new conclusion section and a new paragraph in the discussion.
- Numerous mistakes, typos and formatting errors have been corrected. The source of all cell lines is now mentioned in the supplementary methods section. The references were double checked and the English writing was carefully corrected throughout the manuscript.
We hope that you will now find our revised manuscript suitable for publication.
All authors have agreed to submit this version of the manuscript, and none has declared any conflicts of interest. We thank you in advance for your attention, and we look forward to hearing from you soon.
Yours sincerely,
Zuzana Saidak
Reviewer #1
- Please provide rationale for using dexamethasone (10µM) in PE/CA-PJ34 and PE/CA-PJ41. Please supplement data with lower and higher concentration respectively. Authors would agree on dose-dependent effect which may modulate results and expression profile respectively.
We thank the reviewer for this important remark. We agree with him/her regarding the need to present dose/response data. The corresponding experimental data is now included as a Suppl. Fig. 3 and cited on p.4 line 190 of the revised manuscript.
Of note, the choice of dexamethasone was based on its well-accepted high specificity for the glucocorticoid receptor, compared for example to cortisol, that displays both glucocorticoid receptor and mineralocorticoid receptor-related transcriptional activities (see for example PMID: 10760051). We inserted a statement along this line in the introduction, since this remark was also made by reviewer #2 (p.2 line 97 of the revised manuscript).
- Line 175-176: In both cell lines, PAI-1-specific 174 immunofluorescence labelling was homogeneous across individual cells, suggesting that the regulation was not limited to a subpopulation of cancer cells (data not shown). Its advised to provide immunofluorescence data in supplementary files.
As requested, we have inserted the corresponding immunofluorescence data as Suppl. Fig. 2.
- Authors report, PAI-1 was not detectable in the cell extracts, however, PAI-1 was detectable in the supernatants and we confirmed the increase in PAI-1 expression after dexamethasone treatment. I would recommend presenting expression data (flow analysis) for PAI-1 with and without dexamethasone. Does it mean PAI-1 is undergoing shedding.
The fact that PAI-1 is secreted has long been known, and it has been extensively reported in the literature (for example PMID: 1632457). While we understand that a flow cytometry study would allow for a separate analysis of the cell membrane expression, we are unable to perform this new set of experiments in the short time-frame that the editor granted us.
We also wish to insist on the fact that our study is centred on the transcriptional regulation of the coagulome by glucocorticoids. While it may be possible that glucocorticoids regulate the secretory apparatus of cancer cells, we believe that this aspect would require an entirely new study.
- Please plot independent replicates in each graph for representation and update the figure 2 and Figure 3C.
We have introduced a statement in the figure legend, in order to explain that the corresponding graphs are representative of three independent experiments. Each experiment gave comparable results. However, because of the fold variability, we chose to present only one representative experiment.
- Manuscript is badly formatted, kind request to review the manuscript for throughout for respective fonts and description of text is inappropriate at many places.
We have thoroughly gone through the manuscript and corrected any inconsistencies. Many small mistakes, typos and formatting errors have been corrected. The references were double checked and the English writing was carefully corrected throughout the manuscript.
- I would not agree with manuscript title to study the role in cancer as the data is specifically with respect to OSCC, please update the title with respect to OSCC. Another suggestion is to supplement data with respect to cell lines of thyroid cancer and pancreatic cancer (refer https://www.proteinatlas.org/ENSG00000122861-PLAU/cell+line) with high PLAU expression for the comaprative profile if authors want to float the manuscript broadly with respect to cancers. I would recommend have comparative results from the remaining two cancers as well.
We thank the reviewer for this comment, and we agree with him/her regarding the need to supplement the current set of data with other, pan-cancer data.
As briefly discussed in our point #3, the use of the protein atlas is not ideally suited to examine the transcriptional regulation of the coagulome by glucocorticoids. Instead, in order to address the reviewer’s request, we have used transcriptomic data that were retrieved from the NCI-60 database. The NCI-60 is a freely available database produced and carried by the US National Cancer Institute, with data on 60 human cancer cell lines, including 54 established from human solid tumors of different primary origin (Reinhold et al, 2012). Using this database, we were able to show the existence of a correlation between the transcriptional output of the GR and the expression of the SERPINE1 mRNA (see the new Fig.5). We believe that these new data now support the general relevance of our title. We are grateful to the reviewer for this important suggestion.
Reviewer #2
- The authors present an analysis of the direct impact of glucocorticoids on the coagulome by transcriptional effects. The paper is written and presented to a very high standard, and contains material that is certainly worthy of publication. Some minor points are raised below, but also three major points that could in my view improve the manuscript if addressed.
Minor points : In general the standard of writing and presentation is exemplary: two small issues, line 33 the wrong font has been applied, line 355, the start of the sentence "our study shows..." requires a capital letter
Also, regarding Obradović et al., 2019 (line 104), this is an odd reference bundled with several others to make a generic point that "GR was identified as a modulator of oncogenic signaling and tumor progression, with different and sometimes contrasting effects on tumor growth depending on tumor type and/or stage". In fact, Obradović et al., 2019 make an important point about potential adverse effects of glucocorticoids and would be better referenced in this context.
We are grateful to the reviewer for the positive comments that we have received. The two minor points that the reviewer noted have been addressed in the revised version of the manuscript. The discussion regarding the citation of Obradovic et al. can be found in the point #9 of this letter, on the adverse effects of glucocorticoids.
- Major points.
Choice of OSCC and PAAD: the analysis presented here focuses on Oral Squamous Cell Carcinoma and Pancreatic Adenocarcinoma tumour types. As the manuscript notes, these express the highest levels of F3, PLAU and SERPIN E1. This may imply that these tumours are particularly suited to delivering the findings described here, and applicability to other tumour types should not automatically be assumed. Suggest in the Abstract to write "We addressed the effects of glucocorticoids on the coagulome of human tumors by investigating interactions with Oral Squamous Cell Carcinoma and Pancreatic Adenocarcinoma tumor types." - or something along those lines - (line 31) and also modify the manuscript more generally to avoid any over-reach in going from the specifics of your work to the general.
We agree with the reviewer’s comment, which overlaps with a comment made by reviewer #1 (see point #6 of this letter). As briefly explained earlier, we have inserted new data obtained from the NCI-60 human cell line database. The new data point to the regulation of SERPINE1 by the glucocorticoid receptor in 8 types of primary solid tumors. We have also introduced the suggested change in the abstract. We thank the reviewer for this important comment.
- Adverse effects of glucocorticoids: The manuscript as presented is excellent on the tumor coagulome, but pays insufficient attention to glucocorticoids, which are widely used drugs with many indirect effects. Returning to Obradović et al., 2019, the authors should note that glucocorticoids may promote metastasis in some cancers (the interplay between GR and breast cancer is complex, PMID: 28687451) and has also been associated with some skin cancers via immunosuppressive effects (PMID: 19034275, PMID: 27115293). The potential pleiotropic effects of glucocorticoids should be addressed in the Introduction to provide context to the reader.
Here again, we thank the reviewer for this useful remark, which again partially overlaps with a comment made by reviewer #1. As noted in our point #1 of this letter, the choice of dexamethasone as a widely used drug is now better introduced (see p.2 line 97 of the revised manuscript). We also discuss its high specificity for the glucocorticoid receptor, when compared to cortisol, an endogenous ligand of the GR. Following the reviewer’s suggestion, we now clearly mention the complex and deleterious effects of glucocorticoids. The suggested references were also cited. We are grateful to the reviewer for helping us give a more accurate and precise presentation of the complex effects of glucocorticoids.
- Limitations section: A further way to address these issues would be to introduce an explicit limitations paragraph at the end of Discussion and before Conclusions, to assist the reader in understanding this research in context. This should emphasise the specific nature of this research in examining OSCC and PAAD, review how confident the authors are in extending these conclusions to other tumor types, and highlight that the effects glucocorticoids are not straightforward.
We have inserted a paragraph, where we now explain the limitations of our experimental conclusions. Despite the fact that our experimental observations were obtained with a limited number of primary cancer types, we believe that the key observations are broadly applicable to human cancers, as suggested by the new results using NCI-60. Moreover, we also put emphasis on the need to extend these conclusions in experimental systems using immunocompetent animals with ‘real’ xenografted tumors, in order to fully address the consequences of glucocorticoid signaling on the Tumor immune microenvironment. We agree with the reviewer that this paragraph is important in order to put our research and that of others in perspective.
****************
Thank you again for carefully reading our manuscript and helping us improve our study.
Zuzana Saidak

Reviewer 2 Report
The tumor coagulome as a transcriptional target and a potential effector of glucocorticoids in human cancers
The authors present an analysis of the direct impact of glucocorticoids on the coagulome by transcriptional effects. The paper is written and presented to a very high standard, and contains material that is certainly worthy of publication. Some minor points are raised below, but also three major points that could in my view improve the manuscript if addressed.
Minor points
In general the standard of writing and presentation is exemplary: two small issues, line 33 the wrong font has been applied, line 355, the start of the sentence "our study shows..." requires a capital letter
Also, regarding Obradović et al., 2019 (line 104), this is an odd reference bundled with several others to make a generic point that "GR was identified as a modulator of oncogenic signaling and tumor progression, with different and sometimes contrasting effects on tumor growth depending on tumor type and/or stage". In fact, Obradović et al., 2019 make an important point about potential adverse effects of glucocorticoids and would be better referenced in this context.
Major points
Choice of OSCC and PAAD: the analysis presented here focuses on Oral Squamous Cell Carcinoma and Pancreatic Adenocarcinoma tumour types. As the manuscript notes, these express the highest levels of F3, PLAU and SERPIN E1. This may imply that these tumours are particularly suited to delivering the findings described here, and applicability to other tumour types should not automatically be assumed. Suggest in the Abstract to write "We addressed the effects of glucocorticoids on the coagulome of human tumors by investigating interactions with Oral Squamous Cell Carcinoma and Pancreatic Adenocarcinoma tumor types." - or something along those lines - (line 31) and also modify the manuscript more generally to avoid any over-reach in going from the specifics of your work to the general.
Adverse effects of glucocorticoids: The manuscript as presented is excellent on the tumor coagulome, but pays insufficient attention to glucocorticoids, which are widely used drugs with many indirect effects. Returning to Obradović et al., 2019, the authors should note that glucocorticoids may promote metastasis in some cancers (the interplay between GR and breast cancer is complex, PMID: 28687451) and has also been associated with some skin cancers via immunosuppressive effects (PMID: 19034275, PMID: 27115293). The potential pleiotropic effects of glucocorticoids should be addressed in the Introduction to provide context to the reader.
Limitations section: A further way to address these issues would be to introduce an explicit limitations paragraph at the end of Discussion and before Conclusions, to assist the reader in understanding this research in context. This should emphasise the specific nature of this research in examining OSCC and PAAD, review how confident the authors are in extending these conclusions to other tumor types, and highlight that the effects glucocorticoids are not straightforward.
Author Response

(The authors gave the same response as above.)

Round 2
Reviewer 2 Report
The authors have substantially improved the manuscript and I believe that it is worthy of publication.